# The evaluation of the General Health Questionnaire (GHQ-12) reliability generalization: A meta-analysis

**Ajele Kenni Wojujutari** *, **Erhabor Sunday Idemudia, Lawrence Ejike Ugwu**

Faculty of Humanities, North-West University, Potchefstroom, South Africa

* 54976073@mynwu.ac

## Abstract

### Background

The General Health Questionnaire (GHQ-12) is widely used for detecting psychiatric disorders, but its reliability across different populations remains to be determined.

### Objective

This meta-analysis aims to evaluate the reliability of GHQ-12 across varied cultural and demographic settings.

### Method

This meta-analysis evaluates the reliability of General Health Questionnaire [GHQ-12]' across diverse populations, employing a systematic search strategy and rigorous inclusion criteria. This meta-analysis evaluates the General Health Questionnaire (GHQ-12) using a pre-registered protocol (CRD42023488436) to ensure unbiased results. Data from 20 studies published between 2016–2023 were analysed using a random-effects model, with quality assessment guided by COSMIN Risk of Bias and QUADAS-2. This study enhances our understanding of GHQ-12's psychometric properties.

### Results

For the GHQ-12 subscales, Cronbach's alpha coefficients were 0.72 (90% CI [0.68, 0.75]) for anxiety and depression, 0.82 (90% CI [0.79, 0.86]) for social dysfunction, and 0.72 (90% CI [0.68, 0.76]) for loss of confidence. However, the analysis showed substantial heterogeneity ($I^2 = 90.04\%$), with significant variability in reliability estimates across different studies. The overall Cronbach's alpha was 0.84 (95% CI [0.810, 0.873]) with SE = 0.016 (90% CI [0.68, 0.82], p < .05), indicating moderate to high internal consistency. Quantifying heterogeneity revealed a substantial level (se = 0.0016, $I^2 = 96.7\%$), signifying considerable variability in the reliability estimate among the studies. Results further show Cronbach's alpha coefficients range from 0.82 to 0.85 (95% CI [0.77, 0.86 to 0.81, 0.90]) for the GHQ 12 items.

**Data Availability Statement:** The data supporting the results and conclusions of our study, "The Evaluation General Health Questionnaire (GHQ-12) Reliability Generalization: A Meta-Analysis," are available in the manuscript itself. All relevant data,

including statistical outputs and key findings, can be found in the Results and Tables sections of the manuscript.

**Funding:** The author(s) received no specific funding for this work.

**Competing interests:** The authors have declared that no competing interests exist.

## Conclusion

While reaffirming the GHQ-12's utility in mental health assessment, our findings urge a more cautious and context-aware application of the questionnaire. The substantial heterogeneity and variability in reliability scores indicate a need for further research. Future studies should explore the reasons behind this variability, focusing on cultural, socio-economic, and methodological factors that might influence the GHQ-12's reliability. This critical analysis underscores the need for a deeper understanding of the GHQ-12's applicability and the importance of tailoring mental health assessment tools to specific population characteristics.

## Introduction

Mental health assessment an indispensable pillar in the comprehensive understanding and effective management of psychological well-being. Self-report mental health assessments are commonly used in research contexts, but they have limitations due to the subjective nature of self-reporting and the lack of temporal precision [1, 2]. Amid the myriad assessment tools, the General Health Questionnaire (GHQ-12) has emerged as a versatile instrument designed to appraise mental health status across various populations, cultures, and linguistic contexts [3, 4]. The GHQ-12, rooted in Goldberg and Hillier's theoretical framework, assesses an individual's mental health status through questions encompassing psychological well-being and distress [4].

The GHQ-12 has transcended geographical boundaries, demonstrating its utility across diverse populations, cultures, and languages [5, 6]. Notably, brevity and adaptability have contributed to its widespread application in clinical and research settings, facilitating a quick yet comprehensive screening of psychological distress and well-being [7, 8].

The General Health Questionnaire (GHQ-12) has emerged as a widely utilized instrument for gauging mental health across diverse populations. Studies have shown that individuals with clinically diagnosed emphysema experience poorer general mental health, increased levels of social dysfunction and anhedonia, heightened depression and anxiety, as well as an elevated loss of confidence [9].

The GHQ-12 has been found to have construct and criterion validity, reliability, and gender and age differences among hospitalized patients with COVID-19 [5, 7]. Researchers have used item response theory (IRT) to analyze GHQ-12 responses and have identified subgroups of individuals based on their level of psychological distress [7]. The GHQ-12 has also been used to screen women in the reproductive age group for their mental health status, with results showing that psychological stressors are present in all three groups of women [10]. Additionally, the GHQ-12 has been standardized and used to compare different population cohorts, providing valuable data for public health and clinical practice [11].

The reliability of the General Health Questionnaire-12 (GHQ-12) has been evaluated in various studies. A study that evaluated the reliability and validity of the GHQ-12 in Chinese dental healthcare workers found that the two-factor model of GHQ-12 had good reliability and validity, making it suitable for assessing the mental health status of Chinese dental healthcare workers [12]. The Research investigated the factor structure of the GHQ-12 for South Korean university students and found that Graetz's three-factor model provided the best fit to the data, indicating that the Korean version of the GHQ-12 is a robust measure of general

psychological distress symptoms [13]. A study tested the bifactor structure of the GHQ-12 among Brazilian physicians and found that it had good psychometric properties, although caution is needed when interpreting specific factors [14].

A robust literature review underscores the GHQ-12's extensive usage in various populations worldwide. In a study of South African healthcare workers, the GHQ-12 showed adequate reliability and validity, with a four-factor model suggesting multidimensionality in this group [15]. Another study in Spanish adolescents found that the GHQ-12 is a one-dimensional test for screening psychological distress with excellent psychometric properties [16]. Among male tannery workers in India, the GHQ-12 was found to be reliable and valid, with a three-factor structure [17]. In German and Pakistani university teachers, the GHQ-12 demonstrated good psychometric properties, with high reliability and validity [6, 18]. A Brazilian-Portuguese version of the GHQ-12 showed good internal reliability and factorial validity in both clinical and non-clinical groups [19] Overall, these studies suggest that the GHQ-12 is a reliable and valid tool for measuring mental health in various populations.

The GHQ-12's cross-cultural adaptability has been examined, considering the potential influence of cultural variations on the interpretation of mental health indicators. A study in a Finnish population confirms GHQ-12's reliability for measuring psychological health but suggests limited predictive ability for mental health service use [20]. Similarly, research critically evaluates the GHQ-12 validity for detecting common mental disorders among men in Goa, India, with findings demonstrating acceptable criterion validity but suggests a need for ongoing validation in the local context [21]. A study assessed the measurement equivalence of the GHQ-12 across six ethnic groups in the UK and found the instrument is invariant, allowing valid population comparisons, but caution is urged for individual screenings [22].

In the Spanish version of the GHQ-12, a multidimensional three-factor structure has been identified as the best-fitting model [23]. A study found compelling evidence of robust internal consistency and a stable two-factor structure in the GHQ-12 reliability study among Australian students aged 7–19, affirming its suitability for mental well-being assessment [24]. Research on Korean early childhood teachers establish the GHQ-12 as a dependable tool with a three-factor structure, affirming its reliability and validity for effectively assessing psychiatric symptoms [25].

On contrary, research exploring the factorial validity of the GHQ-12 in outpatients with psychological disorders in China found that the 3-factor model demonstrated superior fit, and the GHQ-12 exhibited measurement invariance across genders, affirming its reliability and applicability in clinical mental health assessments [26]. Similarly, research investigated the factor structures of the GHQ-12 among rural Chinese residents and found adequate reveal reliability of the instrument's two- and three-factor structures, suggesting its applicability for assessing mental health in rural China based on the number of kins [27]. The study emphasises the importance of cultural and contextual considerations in utilising the GHQ-12 in diverse populations. Study validated the Persian version of the GHQ-12 for Iranian elders found the instrument two-factors, demonstrating adequate validity and reliability, making it a suitable tool for assessing the general health of Persian-speaking elderly populations [28].

The study assessed the GHQ-12 reliability for Iranian elders, and found good internal consistency and satisfactory test-retest reliability, supporting its application for a mental health assessment [29]. The study validates the GHQ-12 for assessing psychological distress in chronic low back pain patients' findings demonstrate good reliability, construct validity, and responsiveness, supporting its utility for assessing this population [30]. A study confirms the General Health Questionnaire (GHQ-12) reliability among older adults in India also found high internal consistency and a valid two-factor structure, supporting its applicability [31]. A study evaluated the General Health Questionnaire (GHQ-12) for use with autistic adults

without learning difficulties, and found good psychometric properties, supporting the measure's validity in this population [32]. A study by [33] revealed that the Indonesian adaptation of GHQ-12 demonstrates favourable internal consistency and construct validity. This suggests its appropriateness for mental health screening in primary care patients, acknowledging potential trade-offs between sensitivity and specificity. The GHQ-12 was also, found to have good psychometric properties in autistic adults without learning difficulties [32].

The rationale for employing a meta-analysis lies in its capacity to synthesize findings from numerous studies, allowing for a comprehensive examination of the GHQ-12's reliability. This approach is crucial for capturing the inherent variability in reliability estimates across different populations, periods, and cultural contexts [34]. While previous studies have assessed the psychometric properties of the GHQ-12 [28, 32, 33], a systematic evaluation of its reliability across a many of studies is notably absent. This study seeks to fill this void by conducting a reliability generalization meta-analysis, aiming to identify patterns, trends, and potential moderators that may influence the GHQ-12's reliability.

Findings from this meta-analysis hold substantial implications for mental health practitioners, researchers, and policymakers. A nuanced understanding of the GHQ-12's reliability can guide its appropriate use in diverse settings, ultimately enhancing the accuracy of mental health assessments and the subsequent interventions based on those assessments. In undertaking this meta-analysis, we aspire to contribute valuable insights that refine the understanding of the GHQ-12's reliability and advance the broader field of mental health assessment, aligning with the ongoing efforts to provide efficient and accurate tools for evaluating psychological well-being.

## Methods

### Study design

We conducted a reliability generalization meta-analysis (RG) to comprehensively evaluate the psychometric properties of the General Health Questionnaire (GHQ-12). The study protocol, outlining the specific methods for the reliability generalization meta-analysis, was pre-registered in the Prospero database with the registration number CRD42023488436. This pre-registration was implemented to enhance transparency, prevent selective reporting, and minimize the risk of bias in the study design and analysis.

### Search strategy

This systematic review implemented a predetermined search strategy to comprehensively evaluate the reliability of the General Health Questionnaire (GHQ-12). A thorough search was conducted across multiple databases, including PubMed, PsycINFO, Medline, CHAHL, Science Direct, Scopus, Web of Science, Google Scholar, APA Psycharticle, and EBSCO, targeting studies published between 2016 and 2023 that focused on the GHQ-12's reliability across diverse populations.

The search strategy incorporated relevant keywords such as Reliability, General Health Questionnaire (GHQ-12), Measurement Properties, Assessment, Psychometric properties, Internal consistency, Cronbach's alpha, validity, and reliability. Various combinations of keywords, such as "validity," "Reliability," and "General Health Questionnaire (GHQ-12)," were employed in Boolean search strings, including ("Reliability" OR "psychometric properties") AND ("General Health Questionnaire (GHQ-12)" OR "GHQ-12") AND ("Cronbach's alpha" OR "internal consistency" OR "reliability" OR "validity"). Additionally, TITLE-ABS-KEY("Reliability generalization" OR "Measurement Properties" OR "Assessment" OR "psychometric properties") AND ("General Health Questionnaire (GHQ-12)" OR "GHQ-12") AND

("Measurement Properties" OR "Assessment" OR "Cronbach's alpha" OR "internal consistency" OR "reliability" OR "validity"), and TS = ("Measurement Properties" OR "Assessment" OR "Reliability generalization" OR "psychometric properties") AND TS = ("General Health Questionnaire (GHQ-12)" OR "Measurement Properties" OR "Assessment" OR "CD-RISC-10" OR "CD-RISC-25") AND TS = ("Cronbach's alpha" OR "internal consistency" OR "reliability" OR "validity")) were used. Language restrictions were not imposed, aiming to inclusively incorporate relevant studies on GHQ-12 scales' validity and reliability.

## Inclusion and exclusion criteria

This systematic review encompasses empirical studies that provide quantitative reliability data for the General Health Questionnaire (GHQ-12). The meta-analysis incorporated data from 20 independent studies, selected for their pertinence to the evaluation of psychometric properties of the General Health Questionnaire (GHQ-12).

Inclusion criteria cover cross-sectional and validation studies, along with psychometric evaluations detailing GHQ-12 reliability, specifically focusing on studies reporting reliability metrics such as Cronbach's alpha and test-retest reliability. The inclusion criteria also encompass studies involving participants from diverse age groups (including adolescents, adults, and the elderly) to capture the GHQ-12's applicability across the lifespan. The research spans various cultural and geographic settings, assessing the scale's reliability across different socio-cultural contexts. It includes both clinical populations (individuals diagnosed with mental health disorders) and non-clinical populations (e.g., general community samples, students) to understand the scale's reliability in different mental health states.

The review includes participants of all gender identities and sexual orientations to ensure a comprehensive examination. It considers studies utilising both the original English version of GHQ-12 and validated translations, examining reliability across different linguistic contexts.

Exclusion criteria involve qualitative research, case reports, opinion pieces, theoretical papers without empirical data, systematic reviews, meta-analyses, and non-empirical studies. Non-peer-reviewed materials such as grey literature, dissertations, theses, and conference abstracts without peer review are excluded to maintain scientific rigour.

Additionally, research published in languages other than English is excluded unless a validated translation of GHQ-12 is used, ensuring consistency in the measurement tool's application. By focusing on diverse quantitative designs, this review comprehensively assesses GHQ-12 reliability across different populations and settings, enhancing our understanding of its psychometric robustness and application in measuring psychological well-being. This approach ensures a broad yet precise examination of GHQ-12's empirical utility and generalizability.

## Publication bias and quality assessment

Publication bias was assessed using funnel plots and statistical tests to ensure that the meta-analysis included studies with diverse results, mitigating the risk of bias. In the assessment of study quality, the review method strictly adhered to the reliability of the COnsensus-based Standards for the selection of health Measurement Instruments Risk of Bias checklist (COSMIN RB, [35]). This checklist served as a robust framework, ensuring a thorough evaluation of the methodological quality and potential risk of bias within the studies included in the review.

To enhance the credibility of the assessment, a comprehensive evaluation of study quality was conducted by two independent investigators. This evaluation employed both the Quality Assessment of Diagnostic Accuracy Studies (QUADAS-2, [36] (S1 Fig)) and the COnsensus-based Standards for the selection of health Measurement Instruments Risk of Bias checklist (COSMIN RB, [35] (S1 Table)). This dual and rigorous examination was undertaken to

provide an in-depth analysis of the methodological robustness and potential biases inherent in the studies included in the review. By employing these recognized and validated assessment tools, the quality assessment process aimed to ensure the reliability, transparency, and validity of the findings, ultimately contributing to the credibility of the overall review (see S1 Fig and S1 Table).

## Data extraction and study selection

Systematic extraction of pertinent data from the selected studies involved capturing study characteristics, demographics, reliability coefficients, validity measures, and other essential information related to GHQ-12 psychometric properties. The meta-analysis applied eligibility criteria to select relevant research, with two independent reviewers meticulously screening titles and abstracts. Full-text articles underwent assessment for final inclusion, and to ensure consistency and accuracy, a comprehensive documentation process was implemented. In disagreements, the reviewers reached a consensus, ensuring a robust and reliable compilation of data for further analysis.

## Statistical analysis

A meta-analytic approach was employed to synthesise reliability coefficients across individual studies, utilising random-effects models to address potential heterogeneity. In this meta-analysis, effect sizes were calculated based on Cronbach's alpha, which measures the internal consistency or reliability of a scale.

The formula for effect size is

$$\sqrt{Cronbach's\ alpha\ x\ \left(1 - \frac{1}{Sample\ Size}\right))}$$

A qualitative analysis was summarised and interpreted the reliability of the General Health Questionnaire (GHQ-12). Additionally, a meta-analysis will be conducted, employing a random-effects model to pool reliability coefficients. Heterogeneity was assessed using the $I^2$ statistic and subgroup analyses. Data analysis was performed using the R studio meta for-package.

## Ethical considerations

While formal ethical clearance was not obtained for our study on the reliability generalization of the General Health Questionnaire (GHQ-12) through meta-analysis, our commitment to upholding ethical standards remains unwavering. We prioritize participant confidentiality, secure informed consent, and maintain research integrity. Our dedication to transparency and methodological rigour is evident through PROSPERO protocol registration and adherence to PRISMA guidelines. These practices ensure a thorough and responsible approach, instilling confidence in the scientific community regarding the validity and reliability of our findings.

# Results

## Selection and reliability coefficients induction

Fig 1 illustrates the PRISMA 2020 flowchart outlining the study selection process and screening. The search across various databases, including Medline, APA PsycINFO, CINAHL, APA PSYCHArticle, EBSCO, PubMed, Web of Science, Scopus, ScienceDirect, Google Scholar, and others, yielded a total of 13,572 research articles. The titles and abstracts of these articles were initially screened, excluding 198 duplicated studies.

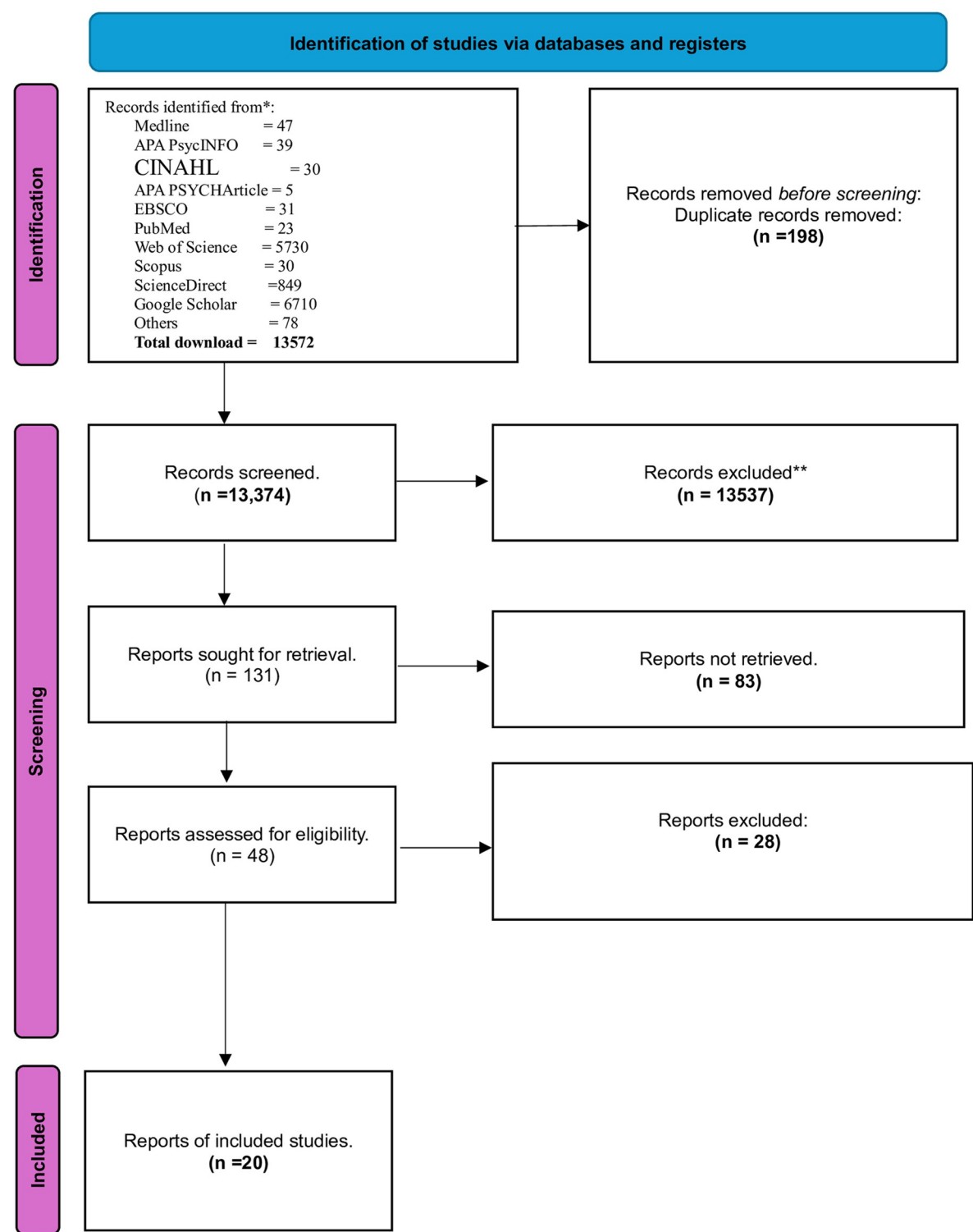

**Fig 1. PRISMA flow chart of study selection for reliability generalization meta-analysis of General Health Questionnaire-12 (GHQ-12).**

During the second screening phase, a total of 13,374 studies were assessed against the inclusion and exclusion criteria, leading to the exclusion of 13,537 articles. This left a pool of 131 studies for further screening. Out of these, only 48 studies were deemed eligible based on the study inclusion criteria. Finally, a subset of 20 studies from the 48 full articles, which met the study inclusion criteria, was included in the meta-analysis.

In examining the psychometric properties of the General Health Questionnaire-12 (GHQ-12) across various studies, diverse patterns emerge, shedding light on the questionnaire's applicability in different contexts. [35] conducted a study in Spain involving 47 participants, female nurses, revealing a robust Cronbach's alpha of 0.872 and a moderately stable test-retest reliability of 0.594. [36] explored GHQ-12 in an Indian undergraduate population [N = 432], finding a Cronbach's alpha of 0.784, though test-retest reliability was not specified. Lee and Kim's investigation [13] in South Korea [N = 504] unveiled a Cronbach's alpha of 0.810 in undergraduate students. Centofanti [24] conducted an extensive study with 18,070 participants in Australia, reporting a Cronbach's alpha of 0.700.

The GHQ-12 exhibited high internal consistency in diverse settings. Elovanio et al.'s [20] study in Finland [N = 4270] with a general population demonstrated a robust Cronbach's alpha of 0.920. Endsley et al. [2017] in India [N = 773] found a Cronbach's alpha of 0.820 among the general population, though conducted in Konkani. The Chinese context was explored by Liu et al. [26] and Guan & Han [27], with N = 870 and N = 32,083, respectively, reporting satisfactory Cronbach's alphas of 0.830 and 0.750. Zhong et al. [2022] investigated dental healthcare workers in China (N = 3,020), revealing a high Cronbach's alpha of 0.892 and a notable test-retest reliability of 0. 843. The synthesis underscores the GHQ-12's reliability across diverse populations and settings, encouraging its use as a valid instrument for mental health assessment. However, careful consideration of cultural and demographic nuances is crucial for nuanced interpretation.

## Overall reliability

The primary aim of this meta-analysis was to evaluate the overall reliability of the General Health Questionnaire (GHQ-12) across a diverse set of studies (k = 20) between 2016 to 2023. The meta-analysis, as summarized in Table 1, demonstrates a robust level of global reliability for the GHQ-12, yielding a coefficient alpha estimate of 0.84 (95% Cl [0.810,0.873]) with SE = 0.016, (90% CI [0.68, 0.82], p < .001), with an individual coefficient alpha estimate for each ranged from 0.70 to 93 (95% Cl [0.69; 0.71 to 0.89; 0.95]) across the 20 included studies. The analysis effect size yielded highly significant (z = 52.05, p < .05), reinforcing the consistent pattern of reliability across diverse contexts of GHQ-12. Quantifying heterogeneity revealed a substantial level, (SE = 0.0016, $I^2$ = 96.7%), signifying considerable variability in reliability estimate among the studies. The result showed a total heterogeneity, $\tau^2$ = 0.0039 (95% Cl [0.0017, 0.0076]), and $\tau$ = 0.0628(95% Cl [0.08418, 0.0869]), as evidenced by the significant Q-test for heterogeneity (Q [df = 19] = 583.73, p < .05). These findings imply that while GHQ-12 maintains high overall reliability, there exists notable heterogeneity in its application across different study settings. Fig 2 provides a visual representation of the meta-analysis results. Also, Fig 3 **provides a visual representation of the included studies bias.**

## GHQ-12 subscale reliability

This study investigates the reliability generalization of the General Health Questionnaire (GHQ-12) subscales, specifically focusing on Anxiety and Depression, Social Dysfunction, and Loss of Confidence. A meta-analysis was conducted to estimate Cronbach's alpha for each subscale, and the results, detailed in Table 1, unveiled varying reliability estimates for individual

**Table 1. Mean reliability coefficients for General Health Questionnaire [GHQ-12].**

| Total scale/Subscales | k | Estimate $\alpha_+$ | Se | Zval | 90%CL | Q | $I^2$ |
|---|---|---|---|---|---|---|---|
| **Coefficient alpha** | | | | | | | |
| **GHQ-12 Global Score** | | | | | | | |
| **Model** | **20** | **0.84** | **0.016** | **52.05** | **0.81[0.87] \*\*\*** | **583.73\*\*\*** | **96.7%** |
| **GHQ-12 Subscales** | | | | | | | |
| Anxiety and Depression | 4 | 0.72 | | | 0.68[0.75] | | |
| Social Dysfunction | 4 | 0.82 | | | 0.79[0.86] | | |
| Loss of confidence | 3 | 0.72 | | | 0.68[0.76] | | |
| **Model** | **3** | **0.75** | **0.04** | **21.03** | **0.68[0.82]\*\*\*** | **20.04\*\*\*** | **90.04%** |
| **GHQ-12 Items** | | | | | | | |
| 1.Lost much sleep | 4 | 0.82 | | | 0.77[0.86] | | |
| 2.Under stress | 4 | 0.84 | | | 0.79[0.88] | | |
| 3.Able to concentrate | 4 | 0.84 | | | 0.80[0.89] | | |
| 4.Playing a useful part | 4 | 0.85 | | | 0.80[0.89] | | |
| 5.Face up to problems | 4 | 0.85 | | | 0.80[0.89] | | |
| 6.Capable of making decisions | 4 | 0.85 | | | 0.81[0.90] | | |
| 7.Could not overcome difficulties | 4 | 0.84 | | | 0.79[0.88] | | |
| 8.Feeling reasonably happy | 4 | 0.84 | | | 0.80[0.89] | | |
| 9.Enjoy your day-to-day activities | 4 | 0.83 | | | 0.78[0.87] | | |
| 10.Feeling unhappy and depressed | 4 | 0.84 | | | 0.80[0.89] | | |
| 11.Losing confidence | 4 | 0.84 | | | 0.79[0.88] | | |
| 12.Thinking of self as worthless | 4 | 0.84 | | | 0.80[0.89] | | |
| **Model** | **12** | **0.84** | **0.007** | **123.56** | **0.83[0.85] \*\*\*** | **1.689** | **0.00%** |
| **Language Version** | | | | | | | |
| Chinese | 2 | 0.79 | | | 0.78[0.80] | | |
| Dutch | 1 | 0.85 | | | 0.71[0.99] | | |
| English | 14 | 0.84 | | | 0.84[0.85] | | |
| Indian-Konkani | 1 | 0.82 | | | 0.76[0.88] | | |
| Iranian-Persian | 1 | 0.82 | | | 0.72[0.92] | | |
| Spanish | 1 | 0.83 | | | 0.80[0.86] | | |
| **Model** | **6** | **0.82** | **0.013** | **61.35** | **0.79[0.85]\*\*\*** | **87.48\*\*\*** | **88.70%** |
| **Test-retest** | | | | | | | |
| **GHQ-12** | | | | | | | |
| **Model** | **4** | **0.78** | | **12.35** | **0.66[0.90]\*\*\*** | **23.63\*\*\*** | **87.3%** |

*Note*: GHQ-12 = General Health Questionnaire (12-items), k = number of studies, $\alpha_+$ = mean coefficient, Q = Cochran's heterogeneity Q statistic, $I^2$ = heterogeneity index

subscales. For Anxiety and Depression, the meta-analysis yielded a Cronbach's alpha coefficient of 0.72 (90% CI [0.68, 0.75]). Similarly, for Social Dysfunction, the estimated Cronbach's alpha was 0.82 (90% CI [0.79, 0.86]), and for Loss of Confidence, it was 0.72 (90% CI [0.68, 0.76]). The overall estimate of Cronbach's Alpha was 0.75 (SE = 0.04, 95% CI [0.68, 0.82]), indicating a moderate to high level of internal consistency across the GHQ-12 scales. The random-effects meta-analysis revealed substantial heterogeneity among the studies ($I^2$ = 90.04%). The Q test for heterogeneity was significant (Q [df = 2] = 20.04, p < .05), indicating notable variability in Cronbach's alpha estimates across the subscales. Fig 4 provides a visual representation of the meta-analysis results.

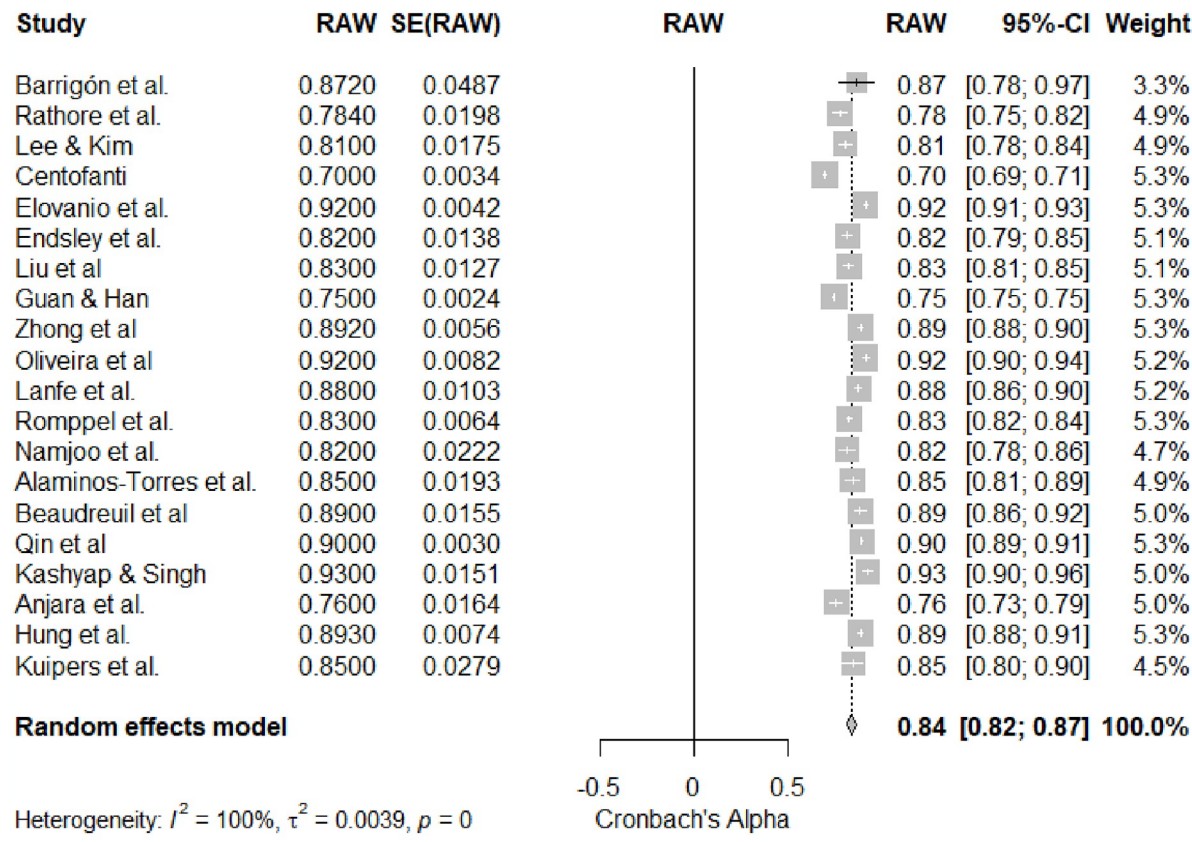

**Fig 2. Forest plot of overall mean reliability coefficients for General Health Questionnaire (GHQ-12).**

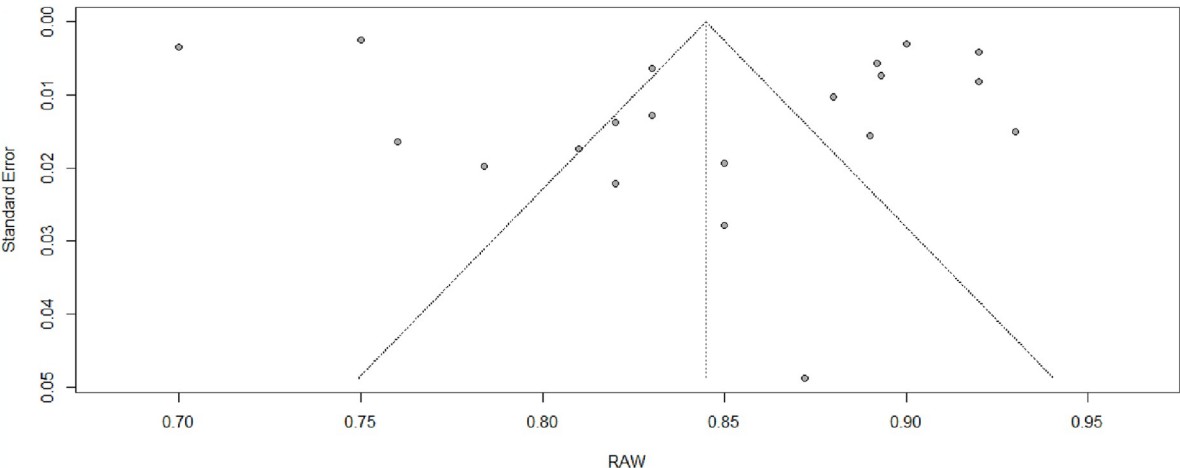

**Fig 3. Funnel plot of overall mean reliability coefficients for General Health Questionnaire (GHQ-12).**

## GHQ-12 items reliability assessment

This analysis aimed to evaluate the reliability of individual items within the General Health Questionnaire (GHQ-12). The specific focus was on the internal consistency, measured by Cronbach's alpha, for each item. Table 1 presents the meta-analysis results for

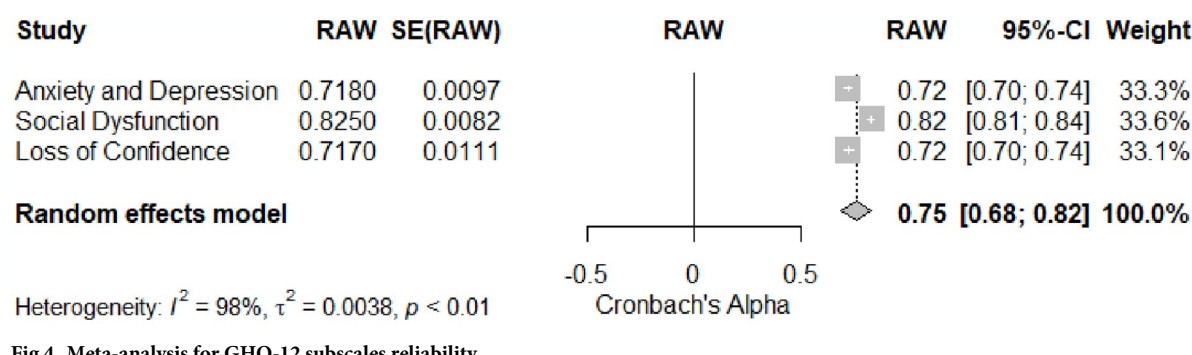

**Fig 4. Meta-analysis for GHQ-12 subscales reliability.**

each GHQ-12 item, including their Cronbach's alpha coefficients and 90% confidence intervals.

Table 1 results showed Lost much sleep (Cronbach's Alpha = 0.82, 90% CI [0.77, 0.86]), Under stress (Cronbach's Alpha = 0.84, 90% CI [0.79, 0.88]), Able to concentrate (Cronbach's Alpha = 0.84, 90% CI [0.80, 0.89]), Playing a useful part (Cronbach's alpha = 0.85, 90% CI [0.80, 0.89]), Face up to problems (Cronbach's Alpha = 0.85, 90% CI [0.80, 0.89]), Capable of making decisions (Cronbach's Alpha = 0.85, 90% CI [0.81, 0.90]), Could not overcome difficulties (Cronbach's Alpha = 0.84, 90% CI [0.79, 0.88]), Feeling reasonably happy (Cronbach's Alpha = 0.84, 90% CI [0.80, 0.89]), Enjoy your day-to-day activities (Cronbach's Alpha = 0.83, 90% CI [0.78, 0.87]), Feeling unhappy and depressed (Cronbach's alpha = 0.84, 90% CI [0.80, 0.89]), Losing confidence (Cronbach's Alpha = 0.84, 90% CI [0.79, 0.88]), and Thinking of self as worthless (Cronbach's Alpha = 0.84, 90% CI [0.80, 0.89]). The overall meta-analysis, combining all GHQ-12 items, yields a Cronbach's alpha of 0.84 (SE = 0.007, 95% CI [0.83, 0.85]). The random-effects meta-analysis revealed substantial heterogeneity among the studies ($I^2$ = 0.00%). The test for heterogeneity is non-significant (Q = 1.689, p> .05), indicating homogeneity in reliability estimates across the 12 items of GHQ.

## Reliability assessment across different language versions of GHQ-12

This analysis aims to explore and compare the reliability variations of the General Health Questionnaire (GHQ-12) across distinct language versions. The primary focus is to evaluate the internal consistency, quantified by Cronbach's Alpha, for each linguistic variant of GHQ-12. Seven language versions are scrutinised, including Chinese, Dutch, English, Indian-Konkani, Iranian Persian, and Spanish. We calculate the sample size, Cronbach's Alpha, and their corresponding 90% confidence intervals for each language iteration, providing a comprehensive overview of the instrument's reliability. Table 1 exhibits the language-dependent analysis, uncovering diverse reliability estimates for individual language versions. Cronbach's alpha coefficients are delineated as follows: 0.79 (90% CL [0.78, 0.80]) for Chinese, 0.85 (90% CL [0.71, 0.99]) for Dutch, 0.84 (90% CL [0.84, 0.85]) for English, 0.82 (90% CL [0.76, 0.88]) for Indian-Konkani, 0.82 (90% CL [0.72, 0.92]) for Iranian Persian, and 0.83 (90% CL [0.80, 0.86]) for Spanish. The collective meta-analysis, encompassing all language versions, yields a Cronbach's Alpha of (M = 0.82) with a 90% confidence interval [0.79, 0.85]. The random-effects meta-analysis revealed substantial heterogeneity among the studies ($I^2$ = 88.70%). Notably, the test for heterogeneity is statistically significant (Q = 87.48, p < .05), indicating variability in reliability estimates across languages.

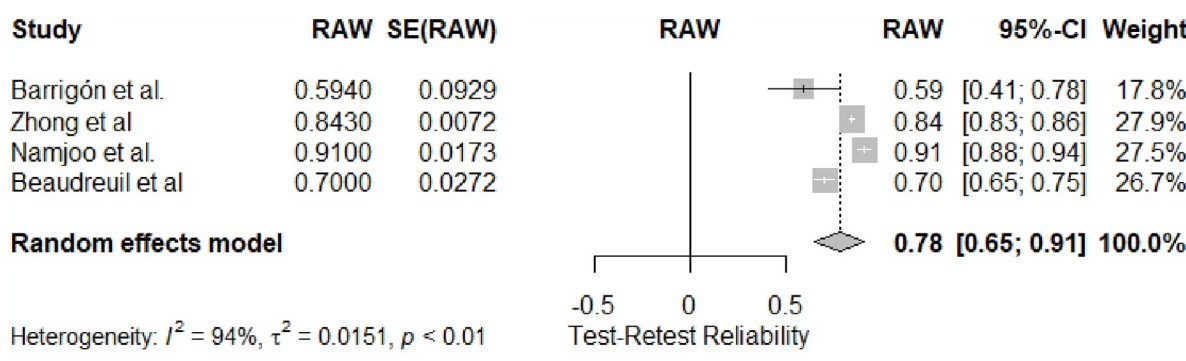

**Fig 5. Forest plot of meta-analysis for GHQ-12 test-retest reliability.**

### GHQ-12 test-retest assessment

The meta-analysis aimed to assess the General Health Questionnaire (GHQ-12) test-retest reliability across a diverse set of studies (k = 4). According to the random-effects model, a significant overall estimate of 0.78 was observed (95% CI: [0.6567, 0.9045]), indicating a moderate to high level of consistency in GHQ-12 test-retest reliability (see Table 1). Regarding heterogeneity, the analysis revealed substantial variability among the studies, with an $I^2$ of 87.3% and a $\tau^2$ of 0.0134. The Q-test for heterogeneity was highly significant (Q = 23.63, df = 3, p < 0.05), indicating significant differences in test-retest reliability estimates across studies. The forest plot (Fig 5) visually depicted individual study estimates and their contributions.

## Discussion

The assessment of the General Health Questionnaire (GHQ-12)'s overall reliability revealed commendable results, backed by a robust coefficient alpha estimate. This aligns with the consensus in existing literature [13, 35], but a critical examination of the methodologies employed in these studies prompts a closer scrutiny of the reliability measures employed. While our findings support GHQ-12's reputation as a reliable mental health assessment tool across diverse populations, it is crucial to question the generalizability of these results and consider potential biases introduced by various study designs.

The second objective focused on investigating the reliability generalization of GHQ-12 subscales. Notably, the Anxiety and Depression subscale exhibited moderate reliability, while the Social Dysfunction and Loss of Confidence subscales demonstrated higher reliability. These results are consistent with previous studies, highlighting variations in reliability across different dimensions of psychological distress [14, 16].

Examining the reliability generalization of GHQ-12 subscales prompts a closer look at the variations observed within the Anxiety and Depression, Social Dysfunction, and Loss of Confidence dimensions. Specifically, the Anxiety and Depression subscale showed moderate reliability, while the Social Dysfunction and Loss of Confidence subscales displayed higher reliability. These findings align with the conclusions drawn by earlier studies, underlining the consistency of reliability variations across diverse dimensions of psychological distress [14, 16]. To gain a more detailed understanding of the instrument's reliability, conducting a more rigorous and critical analysis of the psychometric properties associated with each subscale is imperative.

The evaluation of individual items within GHQ-12, showcasing consistent high reliability, prompts a critical examination of the underlying constructs measured by each item. While the

study echoes [32]' emphasis on robust internal consistency, a deeper exploration into the construct validity of each item is crucial. Critical discussions surrounding the theoretical underpinnings of these items and their alignment with contemporary conceptualizations of mental health will contribute to a more detailed interpretation of the results.

Turning attention to the reliability variations across distinct language versions, our study has revealed noteworthy differences in 'Cronbach's alpha. A critical evaluation of the impact of cultural and linguistic nuances on mental health assessments is imperative. While supporting the findings of [30] and the suitability of the Indonesian version [33], it is crucial to address potential cultural biases and language-related challenges that may affect the cross-cultural applicability of GHQ-12.

Finally, the examination of test-retest reliability has been broadened to underscore the importance of the results obtained from four distinct studies. A critical examination of the potential sources of variability in test-retest reliability is essential, including factors such as the time interval between assessments and the stability of mental health conditions over time. While our results align with [29]'findings among Iranian elders, a more critical analysis will provide a comprehensive understanding of the temporal stability of GHQ-12 scores.

## Implications

This research evaluates the reliability of the General Health Questionnaire (GHQ-12) across diverse populations, highlighting the need for culturally sensitive adaptations. It also emphasises the need for ongoing refinement of mental health assessment tools, guiding future modifications and improvements. The study contributes valuable knowledge to mental health, supporting healthcare professionals and researchers in understanding and addressing mental health issues. It advocates for a combination of qualitative and quantitative approaches to comprehensively understand mental health conditions and their assessment.

## Future directions

Future research on the GHQ-12 should focus on longitudinal studies to better understand its temporal stability and delve into reliability across diverse demographic and cultural groups, particularly underrepresented populations. Emphasis on cross-cultural validation and cultural adaptations of the GHQ-12 will improve its applicability. Detailed examination of subscale reliability and factors influencing variability is crucial, alongside comparisons with standard diagnostic measures. Combining qualitative and quantitative methods will enrich the understanding of mental health assessments. Developing culturally tailored instruments will address cross-cultural variations, enhancing the GHQ-12's effectiveness in diverse settings, thereby refining it as a reliable mental health assessment tool.

## Conclusion

According to our meta-analysis, the General Health Questionnaire (GHQ-12) has good overall reliability, according to our meta-analysis, indicating that it can be used with confidence to assess mental health in a range of populations. Despite variations in subscale dependability, individual items consistently exhibit high reliability. However, the different effects of culture, language, and study heterogeneity emphasise the need for caution when interpreting data. It is a helpful tool that will only improve over time, increasing our understanding of mental health and being more applicable in various contexts.

## Supporting information

**S1 Table. Summary of COSMIN risk of bias [RB] checklist assessments.**
(DOCX)

**S1 Fig. QUADAS-2 assessments for included studies.**
(DOCX)

**S1 Checklist. PRISMA 2020 checklist.**
(DOCX)

## Author Contributions

**Conceptualization:** Ajele Kenni Wojujutari, Lawrence Ejike Ugwu.

**Data curation:** Lawrence Ejike Ugwu.

**Formal analysis:** Lawrence Ejike Ugwu.

**Methodology:** Ajele Kenni Wojujutari, Erhabor Sunday Idemudia.

**Software:** Ajele Kenni Wojujutari.

**Validation:** Ajele Kenni Wojujutari, Erhabor Sunday Idemudia.

**Visualization:** Ajele Kenni Wojujutari.

**Writing – original draft:** Ajele Kenni Wojujutari.

**Writing – review & editing:** Ajele Kenni Wojujutari, Erhabor Sunday Idemudia.

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
