## [Decision Letter · Decision Letter 0]

14 Feb 2024

PONE-D-24-01617The Evaluation General Health Questionnaire (GHQ-12) Reliability Generalization: A Meta-AnalysisPLOS ONE

Dear Dr. Ajele,

Thank you for submitting your manuscript to PLOS ONE. After careful consideration, we feel that it has merit but needs few changes. Therefore, we invite you to submit a revised version of the manuscript that addresses the points raised during the review process.

We look forward to receiving your revised manuscript.

Kind regards,

Shazia Khalid, PhD

Academic Editor

PLOS ONE

Journal Requirements:

"NO authors have competing interests."

Additional Editor Comments:

Dear Author,

The manuscipt is accepted with monir changes, which are given below:

1. The Introduction and Discussion section needs to more detailed with recent references.

2. Inclusion and exclusion criteria should be clearly mentioned.

3. The article needs to be reviewed for grammatical mistakes.

Thank you.

Reviewers' comments:

Reviewer's Responses to Questions

**Comments to the Author**

1. Is the manuscript technically sound, and do the data support the conclusions?

Reviewer #1: Yes

Reviewer #2: Yes

2. Has the statistical analysis been performed appropriately and rigorously? 

Reviewer #1: Yes

Reviewer #2: Yes

3. Have the authors made all data underlying the findings in their manuscript fully available?

Reviewer #1: Yes

Reviewer #2: Yes

4. Is the manuscript presented in an intelligible fashion and written in standard English?

Reviewer #1: Yes

Reviewer #2: Yes

5. Review Comments to the Author

Reviewer #1: Overall this research article seems quite coherent. It covered all of its obejctve clearly. I recommend to make discussion section more argumentatve rather than simply reaching to conlcusions abruptly, authors should build a logcal argument. I think more discussion and analysis is need in discussion section. Make sure toclealry state all point that this study discovered. Moroever there are few typo errors on page 5, 7, 17 & 18. Please recheck. These errors include absence of dot at the end of sentence or other typo errors. Moroever, in sytematic review research we expect it to be more detailed and indepth.This point is lacking in this research specifically in introduction and discsussion section. Please clearly state why you inlcuded certan studies and exlcuded other. Please elaboartae exclusion and inlcusion criteria. Moreover, also mention rational behind it.

Reviewer #2: Overall the research paper impressively and thoroughly explored an essential subject but I would like to share few recommendations.

1. your title should include a word "of" before General health questionnaire as it is creating a little confusion in understanding.

2. Method section of Abstract (4th line) should end with the word "results" (unbiased results) for better understanding.

3. Paper should include a more robust literature review and rationale with reference to the usage of GHQ in diverse populations, cultures and languages.

4. The paper's quality is uplifted by the careful attention given to its methodology regarding the screening procedure but it lacks in understanding of Inclusion criteria. Must add a more clear and specified inclusion criteria regarding diversity of studies. so that it may relate well with the results. Add well explained Procedure in method section.

5. The results section related to Test-Retest reliability is not providing enough information for a good understanding. I would suggest to revise this part.

6. The results regarding "heterogeneity" needs more explanation.

6. PLOS authors have the option to publish the peer review history of their article (what does this mean?). If published, this will include your full peer review and any attached files.

Reviewer #1: **Yes: **Rukhshanda Majeed

Reviewer #2: No

---

## [Author Response · Author response to Decision Letter 0]

20 Feb 2024

Dear Dr. Khalid and Reviewers,

we would like to express my gratitude for the time and effort invested in the thorough evaluation of my manuscript titled "The Evaluation General Health Questionnaire (GHQ-12) Reliability Generalization: A Meta-Analysis" with Manuscript ID PONE-D-24-01617. We appreciate the insightful comments and constructive feedback provided by both the academic editor and the reviewers.

We carefully considered each comment and implemented the necessary revisions to address the concerns raised during the review process. Below is a point-by-point response to the comments and a summary of the changes made:

Reviewer 1

Discussion Section: we expanded the Discussion section, providing more detailed arguments and logical connections to the study's objectives and findings. This includes a comprehensive analysis of the discovered points.

Typos and Aesthetic Presentation: We thoroughly reviewed the manuscript for typographical errors and addressed the identified typos on pages 5, 7, 17, and 18. The overall aesthetic presentation of the systematic review has been improved for clarity.

Inclusion and Exclusion Criteria: We clearly stated the inclusion and exclusion criteria for the systematic review in both the methods and introduction sections. Additionally, I have provided a detailed rationale for the selection of studies, explaining why certain studies were included, and others were excluded.

Reviewer 2 

Title Revision: We revised the title to include the word "of" before "General Health Questionnaire" for better clarity.

Abstract Method Section and Literature Review: The Method section of the abstract has been revised to end with the word "results" for improved understanding. The literature review and rationale have been strengthened by incorporating more robust references related to the usage of GHQ in diverse populations, cultures, and languages.

Inclusion Criteria Explanation: We provided a more clear and specified inclusion criteria section regarding the diversity of studies, ensuring it relates well to the results. A well-explained procedure has been added to the Method section.

Test-Retest Reliability and Heterogeneity: The results related to test-retest reliability have been revised to provide more information for better understanding. The section on heterogeneity has been explained more thoroughly to enhance clarity.

We also addressed the additional journal requirements, including style guidelines, competing interests, data availability statement, ethics statement, supporting information captions, and reference list review. All these changes are reflected in the attached documents: 'Response to Reviewers,' 'Revised Manuscript with Track Changes,' and 'Manuscript.'

We appreciate your guidance and constructive feedback, which have undoubtedly improved the quality of the manuscript. We are confident that the revised version meets the standards for publication in PLOS ONE.

Thank you for your time and consideration. I look forward to any further feedback and the opportunity to proceed with the publication process.

Best regards,

Dr. KW Ajele

---

## [Editor Report · Decision Letter 1]

8 May 2024

The Evaluation of General Health Questionnaire (GHQ-12) Reliability Generalization: A Meta-Analysis

PONE-D-24-01617R1

Dear Autrhor,

We’re pleased to inform you that your manuscript has been judged scientifically suitable for publication and will be formally accepted for publication once it meets all outstanding technical requirements.

Kind regards,

Shazia Khalid, PhD

Academic Editor

PLOS ONE
---

## [Editor Report · Acceptance letter]

8 Jul 2024

PONE-D-24-01617R1 

PLOS ONE

Dear Dr. Wojujutari, 

I'm pleased to inform you that your manuscript has been deemed suitable for publication in PLOS ONE. Congratulations! Your manuscript is now being handed over to our production team.

Kind regards, 

on behalf of

Professor Shazia Khalid 

Academic Editor

PLOS ONE